# Associations between sleep duration and depression, mental health, physical health, and general health in U.S. adults: A population-based study

Mojisola Fasokun[1], Oluwasegun Akinyemi[2], Fadeke Ogunyankin[3], Phiwinhlanhla Ndebele-Ngwenya[2]*, Kaelyn Gordon[2], Seun Ikugbayigbe[4], Uzoamaka Nwosu[5], Mariam Michael[6], Kakra Hughes[7], Temitope Ogundare[8]

**1** Department of Epidemiology, University of Alabama at Birmingham, Birmingham, Alabama, United States of America, **2** The Clive O Callender Outcomes Research Center, Howard University College of Medicine, Washington, District of Columbia, United States of America, **3** Department of Research Data Science and Analytics, Cook Children's Health Care System: Cook Children's Medical Center, Fort Worth, Texas, United States of America, **4** Department of Biological Sciences, Eastern Illinois University, Charleston, Illinois, United States of America, **5** Department of Psychiatry and Behavioral Sciences, Howard University College of Medicine, Washington, District of Columbia, United States of America, **6** Department of Internal Medicine, Howard University College of Medicine, Washington, District of Columbia, United States of America, **7** Department of Surgery, Howard University College of Medicine, Washington, District of Columbia, United States of America, **8** Department of Psychiatry, Boston University School of Medicine, Boston, Massachusetts, United States of America

* phiwinhlanhla.ndebe@bison.howard.edu

## Abstract

### Introduction

Adequate sleep is vital for maintaining mental and physical health. In the United States, a substantial proportion of adults report sleep durations that fall outside the recommended range. Prior research has associated insufficient or excessive sleep with adverse health outcomes; however, few studies have systematically quantified these associations across multiple health indicators using nationally representative data.

### Objective

This study aims to evaluate the impact of short sleep duration on four key health outcomes: depression diagnosis, number of self-reported poor mental health days, number of physically unhealthy days, and self-rated general health status, using nationally representative U.S. data.

### Methodology: Methods

We analyzed nationally representative data from the Behavioral Risk Factor Surveillance System (BRFSS) collected between 2016 and 2023. Sleep duration was

**Data availability statement:** All relevant data are within the manuscript and its supporting information files.

**Funding:** This project was supported (in part) by the National Institute on Minority Health and Health Disparities of the National Institutes of Health under Award Number 2U54MD007597 to W.S. The content is solely the responsibility of the authors and does not necessarily represent the official views of the National Institutes of Health." For clarification, although Dr. William Southerland (W.S.) is not an author on this manuscript, the second author, Dr. Oluwasegun A. Akinyemi, is supported under his NIH center grant. The Article Processing Charge (APC) for this manuscript is being covered through that grant, which is why the funding must be disclosed under Dr. Southerland's award number.

**Competing interests:** The authors have declared that no competing interests exist.

self-reported and categorized into three groups: short sleep (≤5 hours), recommended sleep (6–8 hours), and long sleep (≥9 hours), with short sleep serving as the reference category. The primary health outcomes included: (1) self-reported diagnosis of depression, (2) number of poor mental health days, (3) number of poor physical health days, and (4) self-rated general health, measured on a 5-point Likert scale from excellent to poor. To estimate the effect of sleep duration on these outcomes, we applied Inverse Probability Weighting (IPW) to derive the Average Treatment Effect (ATE), adjusting for key demographic and socioeconomic covariates. All analyses incorporated BRFSS complex survey weights to ensure national representativeness.

## Results

The study included 318,000 adults (63.3% female; 74.5% White) with a mean age of 51.3 ± 18.4 years. Among individuals with recommended sleep duration (6–8 hours), the baseline prevalence of depression was 39.5% (95% CI: 39.4%–39.7%). Compared to this group, short sleep duration (≤5 hours) was associated with a 14.1 percentage point increase in depression incidence (95% CI: 13.8%–14.4%), while long sleep duration (≥9 hours) was linked to a 12.9 percentage point increase (95% CI: 12.5%–13.3%). Those with short sleep reported an average of 5.3 poor mental health days (95% CI: 5.3–5.4), 4.4 poor physical health days (95% CI: 4.3–4.4), and a higher prevalence of poor general health, 10.0% (0.1, 95% CI: 9.7%–10.2%), compared to individuals with recommended sleep. Similarly, individuals with long sleep duration (≥9 hours) also reported more poor mental (4.6 days, 95% CI: 4.5–4.7) and physical health days (3.2 days, 95% CI: 3.1–3.3), along with a higher prevalence of poor general health, 20.3% (20.3%%, 95% CI 19.4%–21.3%) compared to those with recommended sleep.

## Conclusion

Both short (≤5 hours) and long (≥9 hours) sleep durations are significantly associated with increased risk of depression, more days of poor mental and physical health, and worse self-rated general health compared to recommended sleep (6–8 hours). Promoting optimal sleep duration through targeted public health interventions, education, and screening may improve population well-being and reduce sleep-related health disparities.

## Introduction

Sleep is a fundamental biological necessity essential for overall health, cognitive function, and emotional well-being [1,2]. Despite its critical role, sleep deprivation remains a growing public health concern in the United States [3,4]. The Centers for Disease Control and Prevention (CDC) recognizes insufficient sleep as a public health epidemic, with a significant proportion of adults failing to meet the

recommended seven to nine hours of sleep per night [5]. National surveys indicate that approximately one in three U.S. adults experience short sleep duration, with even higher prevalence among racial minorities, low-income populations, and those with demanding work schedules [6–8]. The consequences of insufficient or disrupted sleep extend far beyond personal health, impacting productivity, increasing accident risk, and imposing substantial economic burdens estimated in the billions of dollars annually [9–12]. Furthermore, sleep deprivation is linked to a lower quality of life, heightened risk of chronic illness, and impaired daily functioning, thereby affecting overall well-being [13,14].

Multiple factors contribute to sleep duration variability in the U.S., including occupational stress, socioeconomic conditions, behavioral patterns, and comorbid health conditions [15–17]. Modern lifestyle disruptions such as prolonged screen exposure, inconsistent sleep routines, and psychosocial stress further intensify this trend [18]. Additionally, sleep disturbances commonly co-occur with mental and physical health disorders such as anxiety, depression, and chronic pain, fostering a bidirectional relationship that compounds health risks [19,20]. Reduced sleep is associated with more frequent poor mental health days, greater emotional distress, and impaired physical functioning [21–23]. Consequently, understanding the nuanced relationship between varying levels of sleep and health outcomes is essential to guide effective public health strategies.

While existing literature has documented associations between inadequate sleep and adverse health outcomes [24,25], most studies have treated sleep duration as a binary exposure (sufficient vs. insufficient) and have not comprehensively examined a broader range of health impacts across differentiated sleep categories [26–28]. Moreover, few studies have concurrently assessed the effects of both short (<6 hours) and long (≥9 hours) sleep on a wide spectrum of outcomes, including depression, poor mental and physical health days, and self-rated general health, an important indicator of holistic well-being [29–32]. Recent evidence suggests that both insufficient and excessive sleep durations may be linked to adverse health profiles [33,34], but population-level analyses using representative U.S. data are limited.

To address these gaps, this study investigates the association between sleep duration categorized into three distinct groups: short sleep (≤5 hours), recommended sleep (6–8 hours), and long sleep (≥9 hours) and four self-reported health outcomes: incidence of depression, number of poor mental health days, number of poor physical health days, and general health status. By utilizing a nationally representative dataset and advanced causal modeling techniques, we aim to provide robust, population-level estimates that inform public health policies and clinical practices related to sleep hygiene and health promotion.

## Methodology

### Study design and data source

This study utilizes data from the Behavioral Risk Factor Surveillance System (BRFSS) [35,36], a nationally representative, cross-sectional survey conducted annually by the Centers for Disease Control and Prevention (CDC). The BRFSS collects health-related data from U.S. adults through telephone-based interviews, providing extensive information on health behaviors, chronic conditions, and healthcare access [37]. Data from 2016 to 2023 were used for this study, providing a broad temporal scope to examine trends and associations between sleep duration, mental health outcomes, and physical activity levels. To assess changes in sleep patterns over time, we conducted a trend analysis using BRFSS data from 2016, 2018, 2020, and 2022—the years in which the full sleep module was administered nationally. The analytic sample was derived following the flow diagram presented in Fig 1.

### Primary explanatory variable: Sleep duration

Sleep duration was measured using the BRFSS survey item: "On average, how many hours of sleep do you get in a 24-hour period?" Respondents provided a numeric value from 1 to 24 hours. For analysis, sleep duration was categorized into three groups: Short sleep (≤5 hours) [38,39], Recommended sleep (6–8 hours) [38,40], and Long sleep (≥9 hours)

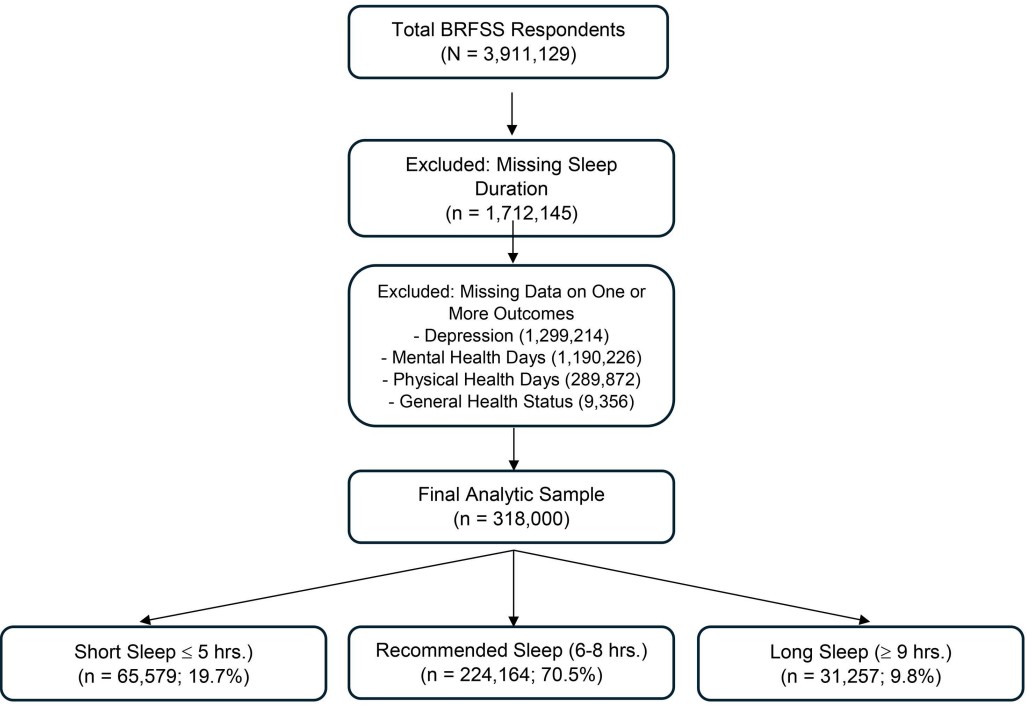

**Fig 1. Study sample derivation flowchart – BRFSS 2016–2023.** This flow diagram illustrates the selection process for the final analytic sample used in the study examining associations between sleep duration and health outcomes. Starting with 3,911,129 adult respondents from the 2016–2023 Behavioral Risk Factor Surveillance System (BRFSS), 1,712,145 participants were excluded due to missing data on sleep duration. An additional 2,788,668 respondents were excluded for missing data on one or more of the four primary outcome variables: depression (n = 1,299,214), poor mental health days (n = 1,190,226), poor physical health days (n = 289,872), and general health status (n = 9,356). The resulting final analytic sample included 318,654 respondents with complete information. These were categorized into: short sleep (≤5 hours, n = 65,579; 19.7%), recommended sleep (6–8 hours, n = 224,164; 70.5%), and long sleep (≥9 hours, n = 31,257; 9.8%).

[38–40]. This categorization was informed by recommendations from the American Academy of Sleep Medicine (AASM) [41,42] and the Centers for Disease Control and Prevention (CDC) [43,44], which endorse 7–9 hours of sleep as optimal for adult health. In all models, short sleep (≤5 hours) served as the reference group, allowing comparisons to both recommended and long sleep durations.

### Outcome variables

The primary outcome variables in this study, drawn from the Behavioral Risk Factor Surveillance System (BRFSS), include self-reported depression diagnosis, poor mental health days, physically unhealthy days, and general health status.

### Depression diagnosis

Depression status was assessed using the BRFSS chronic condition item: "(Ever told) (you had) a depressive disorder (including depression, major depression, dysthymia, or minor depression)?" Responses were coded as 1 = Yes and 2 = No. Individuals responding "Yes" were classified as having a self-reported history of depressive disorder. Non-substantive responses—including "Don't know/Not sure" (code = 7) and "Refused" (code = 9)—were excluded from the analysis. This binary variable was analyzed as a categorical outcome to estimate the association between sleep duration and the likelihood of depression diagnosis.

### Mentally unhealthy days

Mentally unhealthy days were assessed using the BRFSS core question: "Now thinking about your mental health, which includes stress, depression, and problems with emotions, for how many days during the past 30 days was your mental health not good?" Responses ranged from 0 to 30 days, reflecting the respondent's self-reported number of days with poor mental health in the past month. Participants selecting "None" were coded as 0 days, while responses such as "Don't know/Not sure" (77) and "Refused" (99) were treated as missing and excluded from analysis. This variable was modeled as a continuous outcome to quantify the monthly burden of mental distress among respondents.

### Physically unhealthy days

Physically unhealthy days were assessed using the BRFSS core question: "Now thinking about your physical health, which includes physical illness and injury, for how many days during the past 30 days was your physical health not good?" Responses ranged from 0 to 30 days, reflecting the number of days respondents experienced poor physical health in the past month. Responses indicating "None" were coded as 0 days, while non-substantive responses (e.g., "Don't know/Not sure" = 77, "Refused" = 99) were treated as missing and excluded from analysis. This variable was modeled as a continuous outcome to capture variations in physical health burden among participants.

### General health status

Self-rated general health was measured using the standard BRFSS item: "Would you say that in general your health is excellent, very good, good, fair, or poor?" Responses were captured on a five-point Likert-type scale ranging from 1 (excellent) to 5 (poor). For analytical purposes, the variable was treated as a continuous measure, with higher scores indicating poorer self-perceived health. This subjective rating has been widely validated as a reliable indicator of overall health status and a strong predictor of morbidity and mortality in population-based research.

### Covariates and confounder adjustment

To minimize confounding and improve the validity of estimated associations between sleep duration and health outcomes, the analysis adjusted for a comprehensive set of demographic and socioeconomic covariates sourced from the Behavioral Risk Factor Surveillance System (BRFSS). These covariates included:

Age group (18–34, 35–54, and ≥55 years)

Sex (male, female)

Body Mass Index (BMI) category (underweight, normal weight, overweight, obese)

Educational attainment (high school or less, some college, college graduate)

Household income (<$25,000, $25,000–$49,999, $50,000–$99,999, ≥ $100,000)

Marital status (married, previously married, never married)

Covid (pre-covid (2016–2019), post-covid (2020–2023))

These covariates were selected based on prior literature linking them to both sleep behaviors and health outcomes, and to control for potential confounding due to variations in socioeconomic status, lifestyle factors, and demographic characteristics. All models were also adjusted for state of residence and survey year to account for geographic and temporal variation in reporting patterns and policy contexts.

By including these covariates in the analytic models, the study aimed to more accurately isolate the independent association between sleep duration and the primary health outcomes of interest.

## Propensity score adjustment (IPW)

To adjust for potential confounding in estimating the causal effect of sleep duration on various health outcomes, we employed inverse probability weighting (IPW) based on propensity scores. Although the Results section references PSM conceptually, our approach used IPW — a method within the propensity score framework that achieves covariate balance across treatment groups without the need for matched pairs. The rationale for using IPW was to retain the full sample while reducing selection bias due to non-random assignment of sleep duration categories, allowing for estimation of the average treatment effect (ATE).

Propensity scores were estimated via a multinomial logistic regression model, where sleep duration category (short sleep [≤5 hours], recommended sleep [6–8 hours], and long sleep [≥9 hours]) was modeled as a function of observed covariates. The following variables were included in the IPW model based on theoretical and empirical relevance to both sleep patterns and health outcomes: age group, sex (female), race/ethnicity, BMI category, educational attainment, income category, marital status, U.S. state of residence (stfip), and survey year (iyear). The syntax used for IPW estimation was: *teffects ipw (Outcome) (sleep_cat i.age_group3 i.bmi_category i.RACE Femalei.Education i.income4cat i.marital_group3 COVID i.stfip i.iyear) [pweight = _llcpwt].*

We applied BRFSS-provided probability weights (_llcpwt) in all models to ensure national representativeness. Balance diagnostics were conducted by comparing standardized mean differences (SMDs) for covariates before and after weighting. Covariate balance was considered acceptable if SMDs were reduced to below 0.1 following weighting. The use of IPW, rather than traditional matching (e.g., nearest-neighbor), was chosen to avoid sample loss and improve efficiency in a large, complex survey dataset. Robust standard errors were clustered at the state level to account for intra-state correlation.

## Ethical considerations

This study used publicly available, de-identified data from the Behavioral Risk Factor Surveillance System (BRFSS), administered by the Centers for Disease Control and Prevention (CDC). As a secondary analysis of anonymized survey data, this research did not involve direct interaction with human participants and was exempt from Institutional Review Board (IRB) oversight. Informed consent was originally obtained by the CDC during data collection. Given the de-identified and publicly accessible nature of the dataset, no additional consent was required. The study complies with CDC data use policies and adheres to established ethical research standards.

## Statistical analysis

To estimate the association between sleep duration and multiple health outcomes, we employed IPW using treatment-effects models within a counterfactual framework. This approach allows for the estimation of the ATE while accounting for observed confounders that influence both sleep duration and health outcomes.

**Treatment and outcome models.** Sleep duration was operationalized categorically into three treatment groups:

• Short sleep: ≤ 5 hours (reference group),

• Recommended sleep: 6–8 hours,

• Long sleep: ≥ 9 hours.

We implemented a multinomial logistic regression model to predict sleep category membership based on a rich set of covariates. These included:

• Age group (18–34, 35–54, ≥55),

• Sex (male, female),

- Race/ethnicity,

- Body Mass Index (BMI) category (underweight, normal, overweight, obese),

- Education (high school or less, some college, college graduate),

- Household income (<$25K, $25–49K, $50–99K, ≥$100K),

- Marital status (married, previously married, never married),

- State of residence, and

- Survey year.

Inverse probability weights were then generated from these estimated treatment probabilities and applied to model the expected values of each health outcome.

## Estimation of effects

Each health outcome was modeled as the dependent variable in a weighted regression model:

- Depression diagnosis (binary) was analyzed using a weighted linear probability model, allowing for straightforward interpretation of marginal effects as risk differences.

- Poor mental health days, physically unhealthy days, and self-rated general health (continuous) were modeled using weighted mean regressions, assuming normally distributed residuals. These outcomes represent count or ordinal measures with relatively symmetric distributions, justifying use of linear estimators.

All analyses accounted for the complex survey design of BRFSS. We applied the provided BRFSS sampling weights (_llcpwt) to ensure population-level representativeness, and clustered standard errors at the state level to account for intra-state correlation. Fixed effects for state and year were included in all models to control for unobserved regional and temporal heterogeneity.

Statistical significance was assessed at the 0.05 alpha level, and robust standard errors were computed for all estimates. Analyses were conducted using Stata's teffects ipw command, and convergence was confirmed through examination of estimation criteria.

## Sensitivity analysis

To address reviewer concerns regarding the definition of sleep duration, we conducted a sensitivity analysis in which sleep was re-categorized according to American Academy of Sleep Medicine and Centers for Disease Control and Prevention guidance. In this re-specification, short sleep was defined as ≤6 hours, recommended sleep as 7–9 hours, and long sleep as ≥10 hours. We re-estimated all primary models using these revised sleep duration categories. The resulting predicted probabilities and regression estimates were summarized in supplementary tables (S1 File).

## Results

Table 1 presents the sociodemographic and health-related characteristics of the study population stratified by sleep duration categories: short sleep (≤ 5 hours), recommended sleep (6–8 hours), and long sleep (≥9 hours). The total sample included 318,000 individuals. Of these, 65,579 (19.7%) reported sleeping fewer than 6 hours, 224,164 (70.5%) reported sleeping 6–8 hours, and 31,257 (9.8%) reported sleeping more than 8 hours per night.

Age distribution varied significantly across sleep categories. Participants aged 35–54 were most prevalent among short sleepers (37.3%), while long sleepers were predominantly older adults (>55 years, 61.7%). Among those reporting 6–8

**Table 1. Sociodemographic and health characteristics of the study population by sleep duration category (BRFSS 2016-2023).**

| Variables (N = 318,000) | Total Population | Sleep (<5hrs) | Sleep (6–8hrs) | Sleep (>8hrs) | p-value |
|---|---|---|---|---|---|
| | (N = 318,000) | (n = 65,579) (19.7%) | (n = 224,164) (70.5%) | (n = 31,257) (9.8%) | |
| **Age** | | | | | <0.001 |
| 18–34Yr. | 66, 151 (20.7%) | 11,005 (17.6%) | 50,032 (22.3%) | 5,114 (16.0%) | |
| 35–54Yr. | 101,844 (31.9%) | 23,394 (37.3%) | 71,341 (31.8%) | 7,109 (22.3%) | |
| >55Yr. | 150,891 (47.3%) | 28,245 (45.1%) | 102,919 (45.9%) | 19,727 (61.7%) | |
| **Sex** | | | | | <0.001 |
| Male | 117,044 (36.7%) | 23,842 (38.1%) | 82,307 (36.7%) | 10,895 (34.1%) | |
| Female | 201,610 (63.3%) | 38,737 (61.9%) | 141,857 (63.3%) | 21,016 (65.9%) | |
| **BMI** | | | | | <0.001 |
| Underweight | 6,558 (2.2%) | 1,534 (2.7%) | 4,246 (2.1%) | 778 (2.6%) | |
| Normal | 82,030 (28.0%) | 13,680 (23.8%) | 60,318 (29.2%) | 8,032 (27.3%) | |
| Overweight | 89,225 (30.4%) | 16,547 (28.8%) | 64,273 (31.1%) | 8,405 (28.6%) | |
| Obese | 115,661 (39.4%) | 25,622 (44.7%) | 77,815 (37.7%) | 12,224 (41.5%) | |
| **Race/Ethnicity** | | | | | <0.001 |
| White | 225,911 (76.7%) | 40,505 (69.9%) | 161,969 (78.3%) | 23,437 (79.5%) | |
| Black | 25,503 (8.7%) | 7,078 (2.2%) | 15,923 (7.7%) | 2,502 (8.5%) | |
| Hispanic | 16,988 (5.8%) | 3,855 (6.7%) | 11,834 (5.7%) | 1,299 (4.4%) | |
| Other | 26,002 (8.8%) | 6,536 (11.3%) | 17,210 (8.3%) | 2,256 (7.7%) | |
| **Education** | | | | | <0.001 |
| High School or less | 113,071 (35.6%) | 28,076 (45.0%) | 72,092 (32.2%) | 12,903 (40.5%) | |
| Some college | 96,360 (30.3%) | 20,432 (32.7%) | 66,223 (29.6%) | 9,705 (30.5%) | |
| College graduate | 108,587 (34.1%) | 13,929 (22.3%) | 85,409 (38.2%) | 9,249 (29.0%) | |
| **Household Income ($)** | | | | | <0.001 |
| <25K | 50,917 (19.0%) | 15,255 (29.2%) | 29,201 (15.4%) | 6,461 (24.7%) | |
| 25-50K | 62,187 (23.2%) | 14,361 (27.5%) | 40,729 (21.5%) | 7,097 (27.1%) | |
| 50-100K | 67,421 (25.2%) | 11,750 (22.5%) | 49,242 (26.0%) | 6,429 (24.6%) | |
| >100K | 87,182 (32.6%) | 10,818 (20.7%) | 70,163 (37.1%) | 6,201 (23.7%) | |
| **Marital Status** | | | | | <0.001 |
| Married | 149,465 (47.2%) | 24,900 (40.1%) | 111,210 (49.9%) | 13,355 (42.1%) | |
| Previously Married | 97,385 (30.8%) | 23,707 (38.2%) | 61,937 (27.8%) | 11,741 (37.0%) | |
| Never Married | 69,751 (22.0%) | 13,520 (21.8%) | 49,595 (22.3%) | 6,636 (20.9%) | |
| **Covid Era** | | | | | <0.001 |
| pre-Covid | 153,094 (51.5%) | 31,444 (53.6%) | 106,157 (50.8%) | 15,493 (52.0%) | |
| post-Covid | 144,230 (48.5%) | 27,278 (46.5%) | 102,675 (49.2%) | 14,277 (48.0%) | |
| **Outcomes** | | | | | |
| Depression | 150,491 (47.6%) | 37,020 (59.7%) | 95,249 (42.8%) | 18,222 (57.5%) | <0.001 |
| Poor Mental Health days | 12.7 + 10.7 | 17.4 + 11.1 | 11.2 + 10.2 | 14.3 + 11.0 | <0.001 |
| Poor Physical Health days | 12.3 + 11.2 | 16.4 + 11.5 | 10.7 + 10.6 | 15.3 + 11.5 | <0.001 |
| Poor General health days | 3.3 + 1.1 | 3.7 + 1.1 | 3.1 + 1.1 | 3.6 + 1.1 | <0.001 |
| Excellent | 17,331 (5.5%) | 1,873 (3.0%) | 14,216 (6.4%) | 1,242 (3.9%) | <0.001 |
| Very Good | 67,436 (21.2%) | 6,747 (10.8%) | 56,084 (25.1%) | 4,605 (14.5%) | <0.001 |
| Good | 100,540 (31.6%) | 16,143 (25.9%) | 75,848 (33.9%) | 8,549 (26.8%) | <0.001 |
| Fair | 86,257 (27.1%) | 21,750 (34.8%) | 54,393 (24.3%) | 10,114 (11.7%) | <0.001 |
| Poor | 46,614 (14.7%) | 15,935 (25.5%) | 23,335 (10.4%) | 7,344 (23.1%) | <0.001 |

hours of sleep, the majority were aged >55 years (45.9%), followed by those aged 35–54 years (31.8%) and 18–34 years (22.3%).

Sex differences were also significant. Males, who comprised 36.7% of the overall population, were slightly overrepresented in the short sleep group (38.1%) but underrepresented in the long sleep group (34.1%). In contrast, females were slightly overrepresented in the long sleep group (65.9%) compared to their overall proportion (63.3%). These differences, however, were marginal.

Body Mass Index (BMI) categories showed a significant relationship with sleep duration. The proportion of obese individuals was highest among short sleepers (44.7%) and lowest among those with 6–8 hours of sleep (37.7%). Normal weight was most common among those with recommended sleep (29.2%).

Race/ethnicity distribution significantly differed across sleep duration categories. Among individuals reporting short sleep (≤5 hours), 69.9% were White, 2.2% Black, 6.7% Hispanic, and 11.3% Other. In contrast, among those with recommended sleep (6–8 hours), 78.3% were White, 7.7% Black, 5.7% Hispanic, and 8.3% Other. Among long sleepers (>8 hours), 79.5% were White, 8.5% Black, 4.4% Hispanic, and 7.7% Other.

Educational attainment differed substantially by sleep duration. Individuals with a high school education or less—who made up 35.6% of the total population—were notably overrepresented among short sleepers (45.0%). In contrast, college graduates, whose baseline proportion was 34.1%, were slightly overrepresented among those with 6–8 hours of sleep (38.2%), suggesting healthier sleep patterns in this more educated group.

Income was positively associated with sleep duration. Nearly 29.2% of short sleepers reported incomes below $25,000 compared to 15.4% among those with 6–8 hours of sleep. High-income earners (> $100,000) were most concentrated among recommended sleepers (37.1%) and least common among short sleepers (20.7%).

Compared to their baseline proportions—married (47.2%), previously married (30.8%), and never married (22.0%)—previously married individuals were overrepresented among short sleepers (38.2%) but underrepresented among those with recommended sleep (27.8%). Married individuals were slightly overrepresented in the 6–8 hour sleep group (49.9%), suggesting more optimal sleep patterns in this group.

After the onset of the COVID-19 pandemic, subtle shifts in sleep patterns were observed. Compared to their overall representation (48.5%), post-COVID respondents were slightly overrepresented in the recommended sleep group (49.2%) but underrepresented among short sleepers (46.5%) and long sleepers (48.0%). These differences, however, were marginal at best.

Mental and physical health outcomes varied by sleep duration. The prevalence of depression was significantly higher among short sleepers (59.7%) and long sleepers (57.5%) compared to those with recommended sleep (42.8%) ($p < 0.001$). Additionally, poor mental health days were highest among short sleepers (mean $17.4 \pm 11.1$) and long sleepers ($14.3 \pm 11.0$), compared to $11.2 \pm 10.2$ in the recommended group ($p < 0.001$). Similarly, physically unhealthy days followed the same pattern, with short sleepers reporting the highest mean ($16.4 \pm 11.5$), followed by long sleepers ($15.3 \pm 11.5$), and recommended sleepers ($10.7 \pm 10.6$) ($p < 0.001$).

General health status differed by sleep duration (Table 1). Adults reporting recommended sleep had the most favorable profiles, with higher predicted probabilities of excellent (0.077) and very good health (0.258) than those with short (0.049 and 0.197, respectively) or long sleep (0.033 and 0.151, respectively). In contrast, the predicted probabilities of fair and poor health were lowest among those with recommended sleep (0.231 and 0.100) and highest among long sleepers (0.313 and 0.203). These patterns indicate that both short and long sleep durations are associated with worse self-reported health compared with recommended sleep (Table 1).

## Sensitivity analyses

Re-analysis using revised sleep-duration categories (≤6 hours, 7–9 hours, ≥ 10 hours) yielded results consistent with the primary findings (S1 File). The U-shaped association between sleep duration and general health status persisted; both

short and long sleep remained associated with higher predicted probabilities of fair and poor health relative to recommended sleep. Point estimates differed only slightly in magnitude, and overall inference was unchanged.

## Temporal trends in sleep duration (2016–2022)

To assess temporal variation in self-reported sleep duration (Table 2), we conducted a trend analysis using BRFSS data from 2016, 2018, 2020, and 2022—the four years in which the complete sleep module was nationally administered. As shown in Table 2, the prevalence of short sleep duration (≤5 hours per night) modestly declined from 21.5% in 2016 (95% CI: 20.9–22.1%) to 19.8% in 2020 (95% CI: 19.0–20.5%), before increasing slightly to 20.5% in 2022 (95% CI: 19.9–21.1%). In contrast, the proportion of adults reporting recommended sleep duration (6–8 hours) gradually increased across survey waves, rising from 69.5% in 2016 to 71.0% in 2022. Meanwhile, long sleep duration (≥9 hours) remained relatively stable, ranging between 8.5% and 10.0% across the same period. These patterns suggest a modest but consistent shift toward healthier sleep durations over time, providing further context for interpreting the health associations reported in this study (S1 Fig 1 in S1 File).

## Association between sleep duration and incidence of depression

Table 3 presents the estimated Average Treatment Effects (ATE) of sleep duration on the incidence of self-reported depression, using inverse probability weighting (IPW) models. Recommended sleep (6–8 hours) served as the reference group for comparison.

Participants with short sleep duration (≤5 hours) had a significantly higher probability of depression, with an ATE coefficient of 0.141 (Std. Err. = 0.002, z = 90.20, p < 0.001; 95% CI: 0.138–0.144). This indicates a 14.1 percentage-point increase in depression incidence compared to individuals who sleep the recommended number of hours.

Similarly, long sleep duration (≥9 hours) was also associated with a significantly elevated likelihood of depression, with an ATE coefficient of 0.129 (Std. Err. = 0.002, z = 65.99, p < 0.001; 95% CI: 0.125–0.133), representing a 12.9 percentage-point increase compared to the recommended group.

Individuals with recommended sleep had a baseline depression probability of 39.5% (Coef. = 0.395, Std. Err. = 0.0001, z = 581.66, p < 0.001; 95% CI: 0.394–0.397), which was markedly lower than that of both short and long sleepers.

**Table 2. Weighted prevalence of self-reported sleep duration categories across survey years in the United States, BRFSS 2016–2022.**

| Sleep Duration | Year | Proportion | Std. Err. | 95% CI |
|---|---|---|---|---|
| Short Sleep | 2016 | 0.215 | 0.003 | 0.209–0.221 |
| | 2018 | 0.213 | 0.003 | 0.207–0.220 |
| | 2020 | 0.198 | 0.004 | 0.190–0.205 |
| | 2022 | 0.205 | 0.003 | 0.199–0.211 |
| Recommended | 2016 | 0.695 | 0.003 | 0.689–0.702 |
| | 2018 | 0.698 | 0.003 | 0.691–0.705 |
| | 2020 | 0.702 | 0.004 | 0.694–0.711 |
| | 2022 | 0.710 | 0.003 | 0.704–0.717 |
| Long Sleep | 2016 | 0.090 | 0.002 | 0.086–0.094 |
| | 2018 | 0.088 | 0.002 | 0.085–0.092 |
| | 2020 | 0.100 | 0.003 | 0.094–0.106 |
| | 2022 | 0.085 | 0.002 | 0.081–0.089 |

Proportions reflect weighted estimates based on national sampling weights from the Behavioral Risk Factor Surveillance System (BRFSS). Sleep duration was classified as: Short Sleep (≤5 hours), Recommended Sleep (6–8 hours), and Long Sleep (≥9 hours). Estimates are restricted to survey years in which the complete sleep module was administered nationally (even years only).

**Table 3. Average treatment effects of sleep duration on incidence of depression.**

| Outcome | Sleep Categories | Coef. | Std. Err. | z-value | p-value | 95% CI | |
|---------|------------------|-------|-----------|---------|---------|--------|---|
| Depression | Short Sleep | 0.141 | 0.002 | 90.20 | <0.001 | 0.138 | 0.144 |
| | Long Sleep | 0.129 | 0.002 | 65.99 | 0.521 | 0.125 | 0.133 |
| | Recommended | 0.395 | 0.0001 | 581.66 | <0.001 | 0.394 | 0.397 |

This table presents the estimated Average Treatment Effects (ATE) of sleep duration categories on the incidence of self-reported depression, using inverse probability weighting (IPW) models. The reference group is individuals reporting recommended sleep (6–8 hours per night). Coefficients represent the estimated difference in the probability of depression relative to the reference group. Positive values indicate a higher probability of depression compared to recommended sleepers. The model adjusts for demographic and socioeconomic covariates, including age group, sex, race, BMI category, income, education, marital status, state, and survey year. Robust standard errors were used for statistical inference. Reported statistics include the coefficient (Coef.), standard error (Std. Err.), z-value, p-value, and 95% confidence interval (CI).

These findings demonstrate a clear U-shaped association between sleep duration and depression risk, with both insufficient and excessive sleep durations linked to significantly higher odds of depression compared to the optimal 6–8 hours (S1 Fig 2 in S1 File).

### Association between sleep duration and poor mental health days

Table 4 presents the ATE of sleep duration on the number of poor mental health days reported in the past 30 days, using IPW models. Individuals with recommended sleep (6–8 hours) served as the reference group.

Compared to the reference, individuals with short sleep duration (≤5 hours) reported a substantially higher number of poor mental health days, with an ATE coefficient of 5.319 (Std. Err. = 0.035, z = 150.51, p < 0.001; 95% CI: 5.250–5.389). Similarly, those with long sleep duration (≥9 hours) also reported significantly more poor mental health days, though to a lesser degree, with a coefficient of 2.456 (Std. Err. = 0.043, z = 56.72, p < 0.001; 95% CI: 2.371–2.541).

Individuals in the recommended sleep group reported a baseline average of 12.129 poor mental health days per month (Std. Err. = 0.015, z = 800.76, p < 0.001; 95% CI: 12.100–12.159), which was significantly lower than that of short and long sleepers.

These results highlight a clear inverse relationship between sleep duration and mental health burden. Both insufficient (≤5 hours) and excessive (≥9 hours) sleep durations are associated with an increased number of poor mental health days, reinforcing the mental health benefits of maintaining a recommended sleep schedule.

**Table 4. Average treatment effects of sleep duration on incidence of poor mental health days.**

| Outcome | Sleep Categories | Coef. | Std. Err. | z-value | p-value | 95% CI | |
|---------|------------------|-------|-----------|---------|---------|--------|---|
| Depression | Short Sleep | 5.319 | 0.035 | 150.51 | <0.001 | 5.250 | 5.389 |
| | Long Sleep | 2.456 | 0.043 | 56.72 | <0.001 | 2.371 | 2.541 |
| | Recommended | 12.129 | 0.015 | 800.76 | <0.001 | 12.100 | 12.159 |

This table displays the estimated Average Treatment Effects (ATE) of sleep duration categories on the number of self-reported poor mental health days in the past 30 days, calculated using inverse probability weighting (IPW) models. The reference group is individuals reporting recommended sleep (6–8 hours per night). Coefficients represent the estimated mean difference in poor mental health days relative to the reference group. Positive values indicate a higher burden of poor mental health compared to recommended sleepers. The model adjusts for key demographic and socioeconomic covariates, including age group, sex, race/ethnicity, BMI category, income, education, marital status, state, and survey year. Reported values include the coefficient (Coef.), standard error (Std. Err.), z-value, p-value, and 95% confidence interval (CI).

## Association between sleep duration and poor physical health days

Table 5 presents the estimated ATE of sleep duration categories on the number of self-reported physically unhealthy days over the past 30 days, derived using the IPW models. The reference group in this analysis is individuals with recommended sleep duration (6–8 hours per night).

Compared to the recommended group, individuals with short sleep duration (≤5 hours) reported 4.379 additional physically unhealthy days per month (Std. Err. = 0.033, z = 132.91, p < 0.001; 95% CI: 4.314–4.443), indicating a substantial physical health burden associated with insufficient sleep. Similarly, those with long sleep duration (≥9 hours) also experienced significantly more physically unhealthy days than recommended sleepers, with an estimated increase of 3.194 days (Std. Err. = 0.041, z = 77.02, p < 0.001; 95% CI: 3.113–3.276).

Individuals in the recommended sleep group reported an average of 10.723 physically unhealthy days per month (Std. Err. = 0.015, z = 729.34, p < 0.001; 95% CI: 10.694–10.752), which was lower than both short and long sleep groups.

These findings suggest that deviations from recommended sleep—either too little or too much—are associated with a higher burden of physical health issues. The largest adverse effect was observed among short sleepers, reinforcing the importance of maintaining a sleep duration of 6–8 hours for optimal physical health.

## Association between sleep duration and general health

Adjusted predictive margins for general health status varied substantially by sleep duration category (Table 6). Individuals reporting recommended sleep duration demonstrated the most favorable health profile overall. The adjusted probability of reporting excellent health was highest among adults with recommended sleep (0.077; 95% CI, 0.075–0.080), and was markedly lower among those with short sleep (0.049; 95% CI, 0.047–0.051) and long sleep (0.033; 95% CI, 0.031–0.036). A similar pattern was observed for reports of very good health, which were most common among individuals with recommended sleep (0.258; 95% CI, 0.254–0.261), declining among those with short sleep (0.197; 95% CI, 0.194–0.201) and long sleep (0.151; 95% CI, 0.143–0.158).

In contrast, the probability of reporting fair or poor health increased notably with departure from recommended sleep duration. Adults with long sleep demonstrated the highest predicted probability of fair health (0.313; 95% CI, 0.307–0.319), followed by those with short sleep (0.280; 95% CI, 0.276–0.284), and recommended sleepers (0.231; 95% CI, 0.227–0.234). This gradient was more pronounced for poor health: long sleepers exhibited a predicted probability of 0.203 (95% CI, 0.194–0.213), followed by short sleepers (0.149; 95% CI, 0.146–0.152) and recommended sleepers (0.100; 95% CI, s0.097–0.102).

Patterns for good health status were comparatively similar across groups, although recommended and short sleepers demonstrated slightly higher probabilities (0.335; 95% CI, 0.331–0.338 and 0.325; 95% CI, 0.321–0.328, respectively)

**Table 5. Average treatment effects of sleep duration on incidence of poor physical health days.**

| Outcome | Sleep Categories | Coef. | Std. Err. | z-value | p-value | 95% CI | |
|---|---|---|---|---|---|---|---|
| Poor Physical Health days | Short Sleep | 4.379 | 0.033 | 132.91 | <0.001 | 4.314 | 4.443 |
| | Long Sleep | 3.194 | 0.041 | 77.02 | <0.001 | 3.113 | 3.276 |
| | Recommended | 10.723 | 0.015 | 729.34 | <0.001 | 10.694 | 10.752 |

This table reports the estimated Average Treatment Effects (ATE) of sleep duration categories on the number of self-reported physically unhealthy days in the past 30 days, using inverse probability weighting (IPW) models. The reference group is individuals with recommended sleep duration (6–8 hours per night). Coefficients represent the estimated mean difference in physically unhealthy days compared to the recommended sleep group. Positive coefficients indicate a greater number of poor physical health days relative to recommended sleepers. The model was adjusted for demographic and socioeconomic variables, including age, sex, race/ethnicity, BMI, education level, income, marital status, geographic region, and survey year. Reported values include the coefficient (Coef.), standard error (Std. Err.), z-value, p-value, and 95% confidence interval (CI).

**Table 6. Adjusted predicted probabilities of general health categories by sleep duration.**

| GENHLTH | Sleep Category | Predicted Probability | Std. Err. | z-value | 95% CI |
|---|---|---|---|---|---|
| Excellent | Recommended | 0.077 | 0.001 | 64.41 | 0.075–0.080 |
| Excellent | Short | 0.049 | 0.001 | 53.48 | 0.047–0.051 |
| Excellent | Long | 0.033 | 0.001 | 26.68 | 0.031–0.036 |
| Very Good | Recommended | 0.258 | 0.002 | 132.92 | 0.254–0.261 |
| Very Good | Short | 0.197 | 0.002 | 112.67 | 0.194–0.201 |
| Very Good | Long | 0.151 | 0.004 | 40.17 | 0.143–0.158 |
| Good | Recommended | 0.335 | 0.002 | 176.22 | 0.331–0.338 |
| Good | Short | 0.325 | 0.002 | 177.88 | 0.321–0.328 |
| Good | Long | 0.300 | 0.003 | 99.75 | 0.294–0.306 |
| Fair | Recommended | 0.231 | 0.002 | 136.64 | 0.227–0.234 |
| Fair | Short | 0.280 | 0.002 | 145.79 | 0.276–0.284 |
| Fair | Long | 0.313 | 0.003 | 102.38 | 0.307–0.319 |
| Poor | Recommended | 0.100 | 0.001 | 79.22 | 0.097–0.102 |
| Poor | Short | 0.149 | 0.002 | 95.67 | 0.146–0.152 |
| Poor | Long | 0.203 | 0.005 | 40.52 | 0.194–0.213 |

Table presents adjusted predicted probabilities for each general health category by sleep duration derived from IPW models.

than long sleepers (0.300; 95% CI, 0.294–0.306). Collectively, these findings indicate a graded, U-shaped association, wherein both short and long sleep are associated with less favorable self-rated general health compared with recommended sleep duration.

## Discussion

In this large, nationally representative sample of over 300,000 U.S. adults, we observed significant associations between sleep duration and key health outcomes, including depression, poor mental health days, poor physical health days, and self-rated general health. Using IPW to adjust for confounding, we found that both short sleep (≤5 hours) and long sleep (≥9 hours) were independently associated with worse health outcomes compared to individuals who reported recommended sleep duration (6–8 hours).

Our findings reinforce previous literature that short sleep duration is a strong predictor of adverse mental and physical health outcomes [21,45–47]. Adults reporting ≤5 hours of sleep had significantly higher incidence of depression (+14.1%, 95% CI: 13.8%–14.4%), longer duration of poor mental health (5.3 days), and poor physical health (4.4 days) in the past month. Furthermore, 44.8% of short sleepers rated their general health as fair or poor. These outcomes are in line with existing evidence that sleep deprivation can dysregulate emotional [48,49], cognitive [48,50], and immune [48,51,52] functioning.

Interestingly, and consistent with emerging literature [53,54], we also found that long sleep duration (≥9 hours) was associated with higher burden of depression and poorer general health compared to the reference group. Though traditionally underexplored, prolonged sleep has increasingly been implicated in poor health outcomes, potentially reflecting underlying undiagnosed health conditions or behavioral and social determinants of health [55–57]. For example, those in the ≥9-hour group had a 12.9% increase in depression incidence and reported worse mental and physical health than the 6–8-hour group, albeit with slightly lower burden than short sleepers. This U-shaped association highlights that both insufficient and excessive sleep are detrimental to health.

Importantly, we accounted for the potential influence of the COVID-19 pandemic by including survey year and state fixed effects in our analysis. The pandemic period did not significantly modify the observed associations, suggesting that the link between sleep duration and health outcomes persisted regardless of broader societal disruptions.

Moreover, our findings regarding the number of poor mental health days are consistent with national trends [58]. A recent analysis by Udupa et al. (2023) reported an increase in the burden of poor mental health days among U.S. adults, with notable disparities by sex, age, and income [59]. Our study corroborates these trends and extends the evidence by linking these poor mental health days directly to sleep duration [60], further supporting calls for prioritizing sleep health in population-level interventions.

The inclusion of general health as an outcome in this study strengthens its contribution to literature [61–63]. The strong association between suboptimal sleep and self-rated poor general health reflects sleep's critical role in broader wellness perceptions and functional status. These findings emphasize that interventions targeting sleep hygiene may not only mitigate psychiatric and physical morbidity but also improve perceived quality of life.

## Public health implications

Our findings carry substantial public health and policy implications, underscoring the critical role of optimal sleep duration, neither too short nor too long, in promoting mental, physical, and general health. The results reveal a U-shaped association where both short sleep (≤5 hours) and long sleep (≥9 hours) are linked with increased incidence of depression, more mentally and physically unhealthy days, and worse self-rated general health. This suggests that public health messaging should shift from simply promoting "more sleep" to advocating for an optimal range, particularly 6–8 hours, as supported by organizations such as the CDC and the American Academy of Sleep Medicine.

Given the high prevalence of short sleep in the U.S., especially among working adults and socioeconomically disadvantaged populations, routine screening for sleep patterns should be integrated into primary care and behavioral health visits. In addition, the growing body of evidence, reinforced by our findings, points to a need to assess for long sleep as a potential marker of underlying health risk, rather than dismissing it as benign.

Policy interventions could include public education campaigns promoting the 6–8-hour sleep guideline, employer-based initiatives that encourage healthy work–life balance and limit excessive shift work, and insurance coverage for evidence-based sleep interventions like cognitive-behavioral therapy for insomnia (CBT-I). Technology-assisted tools such as sleep tracking apps and behavioral nudges via telehealth may help scale these efforts.

Finally, our findings are consistent with national estimates, including data from Udupa et al. (2023), showing an increase in poor mental health days among U.S. adults. Addressing sleep as a modifiable behavioral target could be a scalable and cost-effective strategy to help reverse these mental health trends, particularly when tailored to high-risk subgroups.

## Strengths and limitations

This study possesses several methodological strengths that enhance its validity, robustness, and generalizability. Leveraging a large, nationally representative sample from the Behavioral Risk Factor Surveillance System (BRFSS), our findings are applicable to diverse adult populations across the United States. The categorization of sleep duration into three distinct groups (short: ≤ 5 hours, recommended: 6–8 hours, and long: ≥ 9 hours) enabled a nuanced assessment of sleep-health relationships beyond the conventional binary classification. Importantly, the application of IPW allowed for adjustment of key sociodemographic and behavioral confounders and strengthened the causal interpretation of the average treatment effects of sleep duration on depression, self-rated general health, and mentally and physically unhealthy days. We also accounted for state and survey year fixed effects, including adjustment for the COVID-19 period, ensuring temporal and regional influences were considered.

Nevertheless, several limitations warrant consideration. First, sleep duration was self-reported, introducing the possibility of recall bias or misclassification. Recent evidence suggests that a proportion of individuals who self-report long sleep

may actually obtain insufficient sleep when measured objectively, indicating potential misclassification among long sleepers and the need for cautious interpretation of this category. Objective sleep assessments, such as actigraphy or polysomnography, would yield more precise measures but were unavailable in this dataset. Second, the cross-sectional design limits causal inference and cannot fully address the bidirectionality between sleep and health outcomes. For instance, while short or long sleep durations may increase the risk of depression or poorer physical health, these conditions may also disrupt sleep patterns, leading to cyclical feedback loops. Third, while we adjusted for a broad array of covariates, residual confounding by unmeasured variables, such as undiagnosed sleep disorders, chronic stress, or genetic predispositions, cannot be entirely ruled out. Lastly, the BRFSS sleep module was administered only in selected years, potentially affecting temporal trend analysis despite our efforts to stratify by survey year.

Future research should employ longitudinal and experimental designs to clarify causality and explore potential biological and behavioral mechanisms underlying the associations observed. Such work may also identify vulnerable subpopulations who would benefit most from targeted sleep health interventions.

## Conclusion

This study provides robust evidence that both short (≤5 hours) and long (≥9 hours) sleep durations are significantly associated with adverse health outcomes compared to the recommended 6–8 hours of sleep. Individuals in the short and long sleep categories exhibited a higher incidence of depression, more days of poor mental and physical health, and worse overall self-rated general health. These findings reinforce the importance of maintaining optimal sleep duration as a key determinant of population mental and physical well-being. Public health efforts should prioritize strategies to promote sleep health, identify at-risk individuals, and address underlying factors contributing to inadequate or excessive sleep. Tailored interventions, health education campaigns, and clinical screening for sleep patterns may collectively improve individual outcomes and reduce the burden of sleep-related health disparities across the United States.

## Supporting information

**S1 File. Supplementary figures and tables.** Contains Supplementary Figure 1 (Trends in Sleep Duration Categories), Supplementary Figure 2 (Prevalence of Depression by Sleep Duration), and Supplementary Tables 1–5 describing baseline characteristics, treatment effects, mentally and physically unhealthy days, and predicted general health probabilities. (DOCX)

## Author contributions

**Conceptualization:** Mojisola Fasokun, Oluwasegun Akinyemi, Fadeke Ogunyankin, Phiwinhlanhla Ndebele-Ngwenya, Kaelyn Gordon, Seun Ikugbayigbe, Uzoamaka Nwosu, Mariam Michael, Kakra Hughes, Temitope Ogundare.

**Data curation:** Mojisola Fasokun, Oluwasegun Akinyemi, Phiwinhlanhla Ndebele-Ngwenya, Uzoamaka Nwosu, Mariam Michael, Temitope Ogundare.

**Formal analysis:** Oluwasegun Akinyemi.

**Funding acquisition:** Oluwasegun Akinyemi, Mariam Michael, Kakra Hughes.

**Investigation:** Mojisola Fasokun, Oluwasegun Akinyemi, Fadeke Ogunyankin, Phiwinhlanhla Ndebele-Ngwenya, Kaelyn Gordon, Seun Ikugbayigbe, Uzoamaka Nwosu, Mariam Michael, Kakra Hughes, Temitope Ogundare.

**Methodology:** Mojisola Fasokun, Oluwasegun Akinyemi, Fadeke Ogunyankin, Phiwinhlanhla Ndebele-Ngwenya, Kaelyn Gordon, Seun Ikugbayigbe, Uzoamaka Nwosu, Mariam Michael, Kakra Hughes, Temitope Ogundare.

**Project administration:** Mojisola Fasokun, Oluwasegun Akinyemi, Fadeke Ogunyankin, Phiwinhlanhla Ndebele-Ngwenya, Kaelyn Gordon, Seun Ikugbayigbe, Uzoamaka Nwosu, Mariam Michael, Kakra Hughes, Temitope Ogundare.

**Resources:** Mojisola Fasokun, Oluwasegun Akinyemi, Fadeke Ogunyankin, Phiwinhlanhla Ndebele-Ngwenya, Kaelyn Gordon, Seun Ikugbayigbe, Uzoamaka Nwosu, Mariam Michael, Kakra Hughes, Temitope Ogundare.

**Software:** Mojisola Fasokun, Oluwasegun Akinyemi, Fadeke Ogunyankin, Phiwinhlanhla Ndebele-Ngwenya, Kaelyn Gordon, Seun Ikugbayigbe, Uzoamaka Nwosu, Mariam Michael, Kakra Hughes, Temitope Ogundare.

**Supervision:** Mojisola Fasokun, Oluwasegun Akinyemi, Fadeke Ogunyankin, Kaelyn Gordon, Uzoamaka Nwosu, Mariam Michael, Kakra Hughes, Temitope Ogundare.

**Validation:** Mojisola Fasokun, Oluwasegun Akinyemi, Fadeke Ogunyankin, Phiwinhlanhla Ndebele-Ngwenya, Kaelyn Gordon, Seun Ikugbayigbe, Uzoamaka Nwosu, Mariam Michael, Kakra Hughes, Temitope Ogundare.

**Visualization:** Mojisola Fasokun, Oluwasegun Akinyemi, Fadeke Ogunyankin, Phiwinhlanhla Ndebele-Ngwenya, Kaelyn Gordon, Seun Ikugbayigbe, Uzoamaka Nwosu, Mariam Michael, Kakra Hughes, Temitope Ogundare.

**Writing – original draft:** Mojisola Fasokun, Oluwasegun Akinyemi, Fadeke Ogunyankin, Phiwinhlanhla Ndebele-Ngwenya, Kaelyn Gordon, Seun Ikugbayigbe, Uzoamaka Nwosu, Mariam Michael, Kakra Hughes, Temitope Ogundare.

**Writing – review & editing:** Mojisola Fasokun, Oluwasegun Akinyemi, Fadeke Ogunyankin, Phiwinhlanhla Ndebele-Ngwenya, Mariam Michael, Kakra Hughes, Temitope Ogundare.

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
