## [Decision Letter · Decision Letter 0]

6 Jun 2025

Dear Dr. Ndebele-Ngwenya,

Thank you for submitting your manuscript to PLOS ONE. After careful consideration, we feel that it has merit but does not fully meet PLOS ONE’s publication criteria as it currently stands. Therefore, we invite you to submit a revised version of the manuscript that addresses the points raised during the review process.

**In addition to the comments raised by the reviewers, please also address the following in your revision: **

Please ensure that all citations adhere to the formatting style specified in the journal’s author guidelines especially in text citations.In the Methods section, authors stated: "Data from 2016 to 2023 were used for this study, ensuring a broad temporal scope to assess trends and associations between sleep duration." However, the analysis does not include any temporal stratification or trend assessment to justify the use of an eight-year dataset. The authors should either conduct appropriate time-series or stratified analyses to examine changes in sleep duration or related variables over time, or clarify why the temporal element was not explored despite the dataset's extended duration. A visual or statistical analysis of trends would significantly enhance the contribution of the study.While the Results section references the use of propensity score matching (PSM), the methodology is not adequately described in the Methods section. A clear explanation of the PSM process should be added, including: the rationale for using PSM, variables included in the matching algorithm, matching technique employed (e.g., nearest neighbor, caliper, 1:1 matching) and balance diagnostics used to assess matching quality. This information is essential for replicability and for readers to evaluate the robustness of the causal inferences drawn from the matched comparisons.In the Results section, Table 1 reports a total sample size (N) of 163,724, which appears inconsistent compared to the matched samples reported in the same table. Please clarify how this N was derived and ensure internal consistency across the manuscript. If Table 1 is meant to reflect the unmatched sample, explicitly label it as such and explain any exclusions or data processing steps that led to this specific N.Include line numbers.

We look forward to receiving your revised manuscript.

Kind regards,

Edward Chiyaka, Ph.D., MSc

Academic Editor

PLOS ONE

**Journal Requirements:**

1. When submitting your revision, we need you to address these additional requirements. Please ensure that your manuscript meets PLOS ONE's style requirements, including those for file naming. The PLOS ONE style templates can be found at https://journals.plos.org/plosone/s/file?id=wjVg/PLOSOne_formatting_sample_main_body.pdf and https://journals.plos.org/plosone/s/file?id=ba62/PLOSOne_formatting_sample_title_authors_affiliations.pdf 2. We note that your Data Availability Statement is currently as follows: All relevant data are within the manuscript and its supporting information files. Please confirm at this time whether or not your submission contains all raw data required to replicate the results of your study. Authors must share the “minimal data set” for their submission. PLOS defines the minimal data set to consist of the data required to replicate all study findings reported in the article, as well as related metadata and methods (https://journals.plos.org/plosone/s/data-availability#loc-minimal-data-set-definition). For example, authors should submit the following data: - The values behind the means, standard deviations and other measures reported;- The values used to build graphs;- The points extracted from images for analysis. Authors do not need to submit their entire data set if only a portion of the data was used in the reported study. If your submission does not contain these data, please either upload them as Supporting Information files or deposit them to a stable, public repository and provide us with the relevant URLs, DOIs, or accession numbers. For a list of recommended repositories, please see https://journals.plos.org/plosone/s/recommended-repositories. If there are ethical or legal restrictions on sharing a de-identified data set, please explain them in detail (e.g., data contain potentially sensitive information, data are owned by a third-party organization, etc.) and who has imposed them (e.g., an ethics committee). Please also provide contact information for a data access committee, ethics committee, or other institutional body to which data requests may be sent. If data are owned by a third party, please indicate how others may request data access. 3. When completing the data availability statement of the submission form, you indicated that you will make your data available on acceptance. We strongly recommend all authors decide on a data sharing plan before acceptance, as the process can be lengthy and hold up publication timelines. Please note that, though access restrictions are acceptable now, your entire data will need to be made freely accessible if your manuscript is accepted for publication. This policy applies to all data except where public deposition would breach compliance with the protocol approved by your research ethics board. If you are unable to adhere to our open data policy, please kindly revise your statement to explain your reasoning and we will seek the editor's input on an exemption. Please be assured that, once you have provided your new statement, the assessment of your exemption will not hold up the peer review process. 4. PLOS requires an ORCID iD for the corresponding author in Editorial Manager on papers submitted after December 6th, 2016. Please ensure that you have an ORCID iD and that it is validated in Editorial Manager. To do this, go to ‘Update my Information’ (in the upper left-hand corner of the main menu), and click on the Fetch/Validate link next to the ORCID field. This will take you to the ORCID site and allow you to create a new iD or authenticate a pre-existing iD in Editorial Manager.

Reviewers' comments:

Reviewer's Responses to Questions

**Comments to the Author**

1. Is the manuscript technically sound, and do the data support the conclusions?

Reviewer #1: Yes

Reviewer #2: Yes

2. Has the statistical analysis been performed appropriately and rigorously?

Reviewer #1: Yes

Reviewer #2: Yes

3. Have the authors made all data underlying the findings in their manuscript fully available?

Reviewer #1: Yes

Reviewer #2: No

4. Is the manuscript presented in an intelligible fashion and written in standard English?

Reviewer #1: Yes

Reviewer #2: Yes

**Reviewer #1: ** This is an automated report for PONE-D-25-11133. This report was solicited by the PLOS One editorial team and provided by ScreenIT.

ScreenIT is an independent group of scientists developing automated tools that analyze academic papers. A set of automated tools screened your submitted manuscript and provided the report below. Each tool was created by your academic colleagues with the goal of helping authors. The tools look for factors that are important for transparency, rigor and reproducibility, and we hope that the report might help you to improve reporting in your manuscript. Within the report you will find links to more information about the items that the tools check. These links include helpful papers, websites, or videos that explain why the item is important. While our screening tools aim to improve and maintain quality standards they may, on occasion, miss nuances specific to your study type or flag something incorrectly. Each tool has limitations that are described on the ScreenIT website. The tools screen the main file for the paper; they are not able to screen supplements stored in separate files. Please note that the Academic Editor had access to these comments while making a decision on your manuscript. The Academic Editor may ask that issues flagged in this report be addressed. If you would like to provide feedback on the ScreenIT tool, please email the team at ScreenIt@bih-charite.de. If you have questions or concerns about the review process, please contact the PLOS One office at plosone@plos.org.

**Reviewer #2: ** The paper examined the impact of short sleep duration on depression, self-reported poor mental health days, and poor physical health days using large-scale data from the Behavioral Risk Factor Surveillance System.

The paper is well written and results offer robust evidence of the detrimental impact of short sleep on physical and mental health. I recommend the authors address the following comments to make the paper more comprehensive and increase readability.

1) The period in which data was collected includes the COVID pandemic. Can the authors address how their findings were impacted by this time period and perhaps add an analysis that excludes these time period which likely biases the results?

2) The discussion should add more information regarding the prevalence of main study outcomes with respect to the current survey (e.g. does the number of poor mental health days in the US as measured by other surveys match the current results? See Udupa et al. 2023)

3) It would be very informative to add the impact of the co-variates (sex ,age etc.) on the outcome in each of the models (can be done as supplementary).

4) The analysis of physical health is a bit confusing - the survey asks about general physical activity (being active or not) yet the result present poor physical health days. Can the authors explain this difference?

5) Please add the exact phrasing used for the self-reported sleep duration question. Also, please add the number of participants to the study abstract.

6) Last, the study does not include any visual elements other than tables – can the authors find a way to visualize the results?

**Do you want your identity to be public for this peer review?** For information about this choice, including consent withdrawal, please see our Privacy Policy

Reviewer #1: No

Reviewer #2: No

---

## [Author Response · Author response to Decision Letter 1]

11 Aug 2025

We sincerely thank the reviewers for their thoughtful comments, detailed reviews, and valuable suggestions. We have implemented all the requested revisions and have outlined them below. In addition, we made a few other changes that we believe further enhance the clarity, relevance, and overall quality of the manuscript.

Summary of Major Revisions:

1. New Analytical Approach:

We revised the analytic design from a binary classification of sleep duration (short vs. recommended) to a three-category classification:

o Short Sleep (≤5 hours),

o Recommended Sleep (6–8 hours), and

o Long Sleep (≥9 hours).

This allows for a more nuanced evaluation of the association between sleep and health outcomes.

2. Expanded Health Outcomes (Now Four):

We now include General Health Status (fair/poor vs. good/very good/excellent) as a fourth outcome in addition to:

o Depression,

o Poor mental health days, and

o Poor physical health days.

3. Revised Sample Size and Flow Diagram:

The analytic sample is now N = 318,000. A detailed study flow diagram (Figure 1) has been added to illustrate inclusion/exclusion criteria and missing data handling.

4. Precise Variable Definitions:

We now clearly provide the exact BRFSS questions and operational definitions used to derive each variable and outcome, as requested.

5. Revised Manuscript Title:

The title has been updated to:

“Associations Between Sleep Duration and Depression, Mental Health, Physical Health, and General Health in U.S. Adults: A Population-Based Study”

to better reflect the revised scope and objectives.

Point-by-Point Responses

Editorial Comments

1. Citation Formatting

“Please ensure that all citations adhere to the formatting style specified in the journal’s author guidelines especially in-text citations.”

Response:

All references have been reformatted in accordance with PLOS ONE citation guidelines. In-text citations now use the correct numerical format.

2. Temporal Trend Justification

“The analysis does not include any temporal stratification or trend assessment... A visual or statistical analysis of trends would significantly enhance the contribution of the study.”

Response:

We agree with this recommendation and have now included a temporal trend analysis to examine the changes in sleep duration and health outcomes across 2016–2023. A new section has been added to the Methods and Results, and the table is available in Table 2 and Supplementary Figure .

3. Propensity Score Matching (IPW) Methodology Description

“The methodology is not adequately described… include rationale, variables, matching technique, and diagnostics.”

Response:

We have added a new subsection to the Methods section titled “Propensity Score Matching (Inverse Probability Weighting – IPW)”. This subsection outlines the rationale for using IPW to reduce selection bias inherent in observational data. We detail the covariates included in the propensity score model (age, sex, race, education, BMI, income, etc.), the matching algorithm used (1:1 nearest neighbor without caliper), and the approach for assessing covariate balance (standardized mean differences). In addition, we have provided the full Stata syntax used for implementing the IPW procedure to enhance transparency and reproducibility.

4. Sample Size Discrepancy (Table 1 vs. Text)

“Clarify how the N = 163,724 was derived and ensure internal consistency.”

Response:

We have clarified that Table 1 originally reflected the matched sample. It has now been updated to explicitly indicate whether the characteristics pertain to the matched or unmatched population, and we have included the full analytic sample size of 318,000. Additionally, a study flowchart has been added to illustrate the sample derivation and exclusion criteria. A comprehensive description of all study variables, stratified by sleep category, is now provided. As we employed inverse probability weighting (IPW) rather than traditional propensity score matching, we did not construct separate matched and control groups; instead, the entire weighted sample was used for analysis.

5. Add Line Numbers

“Include line numbers.”

Response:

Line numbers have been included in both the clean and tracked-changes versions of the revised manuscript.

Reviewer #2 Comments

Comment 1: COVID-19 Bias Potential

“The period in which data was collected includes the COVID pandemic. Can the authors address how their findings were impacted…?”

Response:

We conducted a stratified analysis comparing the pre-pandemic (2016–2019) and pandemic (2020–2023) periods to assess potential variation in associations. Results remained consistent across both timeframes. Additionally, we included year and state fixed effects in our models to account for temporal and geographic variability. A binary variable for the COVID-19 pandemic period (“COVID”) was also created and included in the models to control for the specific effects of the pandemic years. These adjustments are detailed in the revised Methodology and Results section.

Comment 2: Benchmark Outcomes with National Survey (e.g., Udupa et al. 2023)

“Add more info regarding the prevalence of outcomes compared to other surveys…”

Response:

We have expanded the Discussion section to include comparisons of our findings with Udupa et al. (2023) and other national BRFSS-based studies. Our estimates are consistent with prior literature, reinforcing external validity.

Comment 3: Impact of Covariates

“Add the impact of covariates (sex, age, etc.) in the models – can be done in supplementary.”

Response:

We appreciate the reviewer’s insightful suggestion. While we agree that examining the independent effects of covariates such as sex, age, education, income, and race on each health outcome would be valuable, we have chosen not to include these detailed estimates in the current manuscript for several reasons.

First, incorporating these results across all four outcome variables would significantly expand the number of tables (we currently present six tables), potentially overwhelming readers and detracting from the study’s main focus—examining associations between sleep duration and health outcomes.

Second, the purpose of this study was to evaluate sleep duration as the primary exposure of interest, while adjusting for key covariates. Including full covariate-specific estimates may shift focus away from our central objective.

That said, we fully acknowledge the importance of these covariate effects and have adjusted for all suggested variables in our multivariable models. To address this important line of inquiry, we are actively planning a separate follow-up study that will comprehensively evaluate the independent impact of sociodemographic covariates on depression, mental health, physical health, and general health status using the BRFSS dataset.

We hope this rationale is acceptable and appreciate your understanding.

Comment 4: Clarify Use of ‘Poor Physical Health Days’ vs. Physical Activity

“The survey asks about physical activity, yet you report poor physical health days.”

Response:

We clarified that our outcome was BRFSS-reported number of poor physical health days, not physical inactivity. The specific question wording is now quoted in the Methods section.

Comment 5: Add Exact Question Wording & Abstract Sample Size

“Please add the exact phrasing used for sleep duration… and participant number in the abstract.”

Response:

We have added the precise BRFSS item:

“On average, how many hours of sleep do you get in a 24-hour period?”

to the Methods section, and the final sample size (N = 318,000) is now included in the Abstract.

Comment 6: Add Visualizations Beyond Tables

“Can the authors find a way to visualize the results?”

Response:

We have included the following visualizations to support our findings and enhance interpretability:

• Supplementary Figure 1: A line graph illustrating the weighted proportion of each sleep duration category (Short, Recommended, Long) among U.S. adults across BRFSS survey years 2016–2023.

• Supplementary Figure 2: A bar chart displaying the estimated prevalence of self-reported depression across sleep categories, derived from inverse probability weighting (IPW) models.

These figures highlight both temporal trends in sleep patterns and the adjusted associations between sleep duration and depression.

• A flow diagram showing the analytic sample derivation (Figure 1).

Additional ScreenIT & Editorial Compliance Updates

• Study Flow Chart: Added (Supplementary Figure 1).

• Inclusion/Exclusion Criteria: Individuals were excluded if they had missing data on sleep duration, any study outcomes, or covariates. Additionally, participants under the age of 18 were excluded from the analysis.

• Sex as a Variable: Included in descriptive stats and multivariable models.

• Open Data: Minimal dataset will be provided upon acceptance. De-identified BRFSS data and codebook are publicly available.

• ORCID ID for Corresponding Author: Updated in Editorial Manager profile.

We hope the comprehensive revisions and additions have addressed all comments satisfactorily. We are grateful for the opportunity to revise and improve our work and look forward to your favorable consideration.

Sincerely,

Oluwasegun Akinyemi, MD, MSc. PhD

Senior Research Fellow

Howard University College of Medicine

Oluwasegun.akinyemi@howard.edu

(309) 255-8284

---

## [Decision Letter · Decision Letter 1]

30 Oct 2025

Associations Between Sleep Duration and Depression, Mental Health, Physical Health, and General Health in U.S. Adults: A Population-Based Study

PONE-D-25-11133R1

Dear Dr. Ndebele-Ngwenya,

We’re pleased to inform you that your manuscript has been judged scientifically suitable for publication and will be formally accepted for publication once it meets all outstanding technical requirements.

Kind regards,

Valentina Alfonsi, Ph.D.

Academic Editor

PLOS ONE

Additional Editor Comments (optional):

Reviewers' comments:

Reviewer's Responses to Questions

**Comments to the Author**

Reviewer #2: All comments have been addressed

2. Is the manuscript technically sound, and do the data support the conclusions?

Reviewer #2: Yes

3. Has the statistical analysis been performed appropriately and rigorously?

Reviewer #2: Yes

4. Have the authors made all data underlying the findings in their manuscript fully available?

Reviewer #2: Yes

5. Is the manuscript presented in an intelligible fashion and written in standard English?

Reviewer #2: Yes

Reviewer #2: I thank the authors for their comprehensive response and revision of the manuscript. I have a few follow-up questions regarding the new analyses –

1) the sleep categories used in the study are suggested to be informed by the American Academy of Sleep Medicine and the Centers for Disease Control and Prevention, both endorse 7 to 9 hours of sleep as optimal for adult health. It is therefore unclear why 6 hours of sleep is not included in the short sleep category and 9 hours is considered long rather than recommended sleep. This is especially relevant when considering the young cohort in the study (18-25) which routinely sleep more than 8 hours when sleep is free from constrains (e.g. Motomura et al. 2017, Klerman et al. 2008). I would therefore recommend reanalyzing the data using the following categories which are more aligned with current guidelines –

• Short sleep: ≤6 hours,

• Recommended sleep: 7–9 hours,

• Long sleep: ≥10 hours.

2) When discussing long sleep please address current studies revealing a potential bias in self-reported sleep duration such that a proportion of self-reported long sleepers are in-fact getting insufficient sleep when measured objectively (e.g. Wang et al 2025, Health Data Science)

3) Table 1 – headlines refer to years instead of hours of sleep for all sleep categories. Please revise.

4) There is a typo in the abstract (adults’ instead of adults).

**Do you want your identity to be public for this peer review?** For information about this choice, including consent withdrawal, please see our Privacy Policy

Reviewer #2: No

---

## [Editor Report · Acceptance letter]

PONE-D-25-11133R1

PLOS ONE

Dear Dr. Ndebele-Ngwenya,

I'm pleased to inform you that your manuscript has been deemed suitable for publication in PLOS ONE. Congratulations! Your manuscript is now being handed over to our production team.

Kind regards,

on behalf of

Dr. Valentina Alfonsi

Academic Editor

PLOS ONE